# Esterification of Lignin Isolated by Deep Eutectic Solvent Using Fatty Acid Chloride, and Its Composite Film with Poly(lactic acid)

**DOI:** 10.3390/polym13132149

**Published:** 2021-06-29

**Authors:** Chan-Woo Park, Song-Yi Han, Rajkumar Bandi, Ramakrishna Dadigala, Eun-Ah Lee, Jeong-Ki Kim, Azelia Wulan Cindradewi, Gu-Joong Kwon, Seung-Hwan Lee

**Affiliations:** 1Institute of Forest Science, Kangwon National University, Chuncheon 24341, Korea; chanwoo8973@kangwon.ac.kr (C.-W.P.); songyi618@kangwon.ac.kr (S.-Y.H.); rajkumar.pgcb@gmail.com (R.B.); dadigala.ramakrishna@gmail.com (R.D.); gjkwon@kangwon.ac.kr (G.-J.K.); 2Department of Forest Biomaterials Engineering, Kangwon National University, Chuncheon 24341, Korea; laa3158@kangwon.ac.kr (E.-A.L.); panda20@kangwon.ac.kr (J.-K.K.); azeliacindradewi@gmail.com (A.W.C.); 3Kangwon Institute of Inclusion Technology, Kangwon National University, Chuncheon 24341, Korea

**Keywords:** lignin, esterification, fatty acid chloride, poly(lactic acid), composite

## Abstract

In this study, the effect of lignin esterification with fatty acid chloride on the properties of lignin and lignin/poly(lactic acid) (PLA) composites was investigated. Lignocellulose (*Pinus densiflora* S. et Z.) was treated using a deep eutectic solvent (DES) with choline chloride (ChCl)/lactic acid (LA). From the DES-soluble fraction, DES-lignin (DL) was isolated by a regeneration process. Lignin esterification was conducted with palmitoyl chloride (PC). As the PC loading increased for DL esterification, the Mw of esterified DL (EDL) was increased, and the glass transition temperature (*T*_g_) was decreased. In DL or EDL/PLA composite films, it was observed that EDL/PLA had cleaner and smoother morphological characteristics than DL/PLA. The addition of DL or EDL in a PLA matrix resulted in a deterioration of tensile properties as compared with neat PLA. The EDL/PLA composite film had a higher tensile strength and elastic modulus than the DL/PLA composite film. DL esterification decreased water absorption with lower water diffusion coefficients. The effect of lignin esterification on improving the compatibility of lignin and PLA was demonstrated. These results are expected to contribute to the development of high-strength lignin composites.

## 1. Introduction

Lignin is the most abundant aromatic biopolymer on earth and accounts for 20–40% of lignocellulose. It consists of three basic structural phenylpropane units: p-coumaryl alcohol, coniferyl alcohol, and sinapyl alcohol [1,2]. Phenylpropane units are randomly cross-linked by linkages of β-O-4, β-5, β-β, and others [1,2]. It has some advantages including biodegradability, biocompatibility, nontoxicity, low density, high carbon content, thermal resistance and oxidation resistance [1,2,3]. Due to these properties, it has great potential utilization in bioplastics, films, carbon fibers, various chemicals, and so on [3,4,5,6,7]. Lignin is mainly produced from black liquor, which is a by-product of the pulping process through a regeneration process by precipitation [8]. Its characteristics differ depending on the type of lignocellulose and the chemical treatment process. During the regeneration of lignin, the depolymerized lignin fragments are recondensed [9,10], which can lead to the formation of recalcitrant condensed units and structural heterogeneity [11]. These heterogeneous structures and characteristics can limit its potential for further application in bioplastics.

A deep eutectic solvent (DES) is a type of ionic liquid composed of molten salts with organic cations and anions with a eutectic point [12,13]. It can be synthesized facilely by mixing hydrogen bond acceptors (HBAs) such as choline chloride (ChCl), betaine, and hydrogen bond donors (HBDs) including lactic acid (LA), oxalic acid and glycerol [12,13,14,15]. DES can offer various advantages, such as facile preparation at low cost, low vapor pressure, nonflammability, sustainability, biodegradability and biocompatibility [15,16]. Because DES can efficiently dissolve the constituents of lignocellulose, especially lignin, they have recently attracted attention as a method for isolating lignin in an ecofriendly way. Moreover, it is known that lignin isolated by DES treatment has a higher hydroxyl group content without sulfur groups, and a molecular structure similar to proto lignin [17,18,19]. Therefore, since DES-lignin (DL) can be easy to chemically modify, it is expected to have excellent usability as various composite materials, including bioplastics.

However, it is well known that the existence of lignin can deteriorate the mechanical properties of plastic composites due to the low compatibility between lignin and other thermoplastic polymers including PLA, polybutylene succinate and polycaprolactone [15,20,21]. To overcome this drawback, improvement in the interaction between lignin and other matrix polymers is highly important. The esterification of lignin can ensure better compatibility with various thermoplastic polymers [22,23,24], resulting in the improvement of lignin dispersity and its mechanical properties in composites [25,26,27]. In addition, esterification can increase lignin mobility, consequently resulting in lower *T*_g_. The decreased *T*_g_ of lignin can increase its thermal flowability, thereby contributing to the improvement of the processability of lignin-based composites with thermoplastic polymers [28]. Esterification of lignin using fatty acids is expected to greatly contribute to the development of lignin-based high-strength composite materials by increasing the processability of lignin and compatibility with various polymers.

In this study, lignin esterification was conducted to improve the physico-mechanical properties of lignin-based composites. Lignin was extracted by DES treatment with ChCl/LA, and the obtained DL was esterified with fatty acid chloride. The esterified DES-lignin (EDL)/PLA composite films were prepared via the solvent casting method. The effect of esterification of lignin on morphological and mechanical properties, as well as thermal and moisture stability in the composite films was investigated.

## 2. Materials and Methods

### 2.1. Materials

Lignocellulose (*Pinus densiflora* S. et Z.) with a lignin content of 31.4% was used as the raw material for lignin isolation. ChCl, LA, pyridine, dimethylformamide (DMF), tetrahydrofuran (THF), and 1M hydrochloric acid solution were purchased from Daejung Chemical & Metals (Siheung, Korea). Palmitoyl chloride (PC) was purchased from Tokyo Chemical Industry (Tokyo, Japan). PLA (IngeoTM Biopolymers 6400D) was obtained from NatureWorks LLC (Minnetonka, MN, USA).

### 2.2. Lignin Isolation

ChCl and LA were mixed in molar ratios of 1/1. The mixture was stirred at 80 °C until it became a clear liquid. The lignocellulose was added into DES with ChCl/LA at a 2 wt% concentration and stirred at 400 rpm at 130 °C for 24 h. The reactant was centrifuged at 4000× *g* for 20 min; thereafter, the DES-soluble fraction and DES-insoluble residue were separated. The DES was washed completely from the DES-insoluble residue by vacuum filtration with a 1,4-dioxane/water (4/1) solution. The filtrated DES with 1,4-dioxane/water was evaporated using a rotary evaporator to remove 1,4-dioxane and water and poured into the DES-soluble fraction obtained by centrifugation. The DES-soluble fraction was dropped in water, and the pH was adjusted to 2.0 by adding 1M HCl solution to regenerate lignin. After stirring for 24 h, the regenerated DL was vacuum-filtrated with distilled water to wash it and freeze-dried for 24 h at −55 °C using a freeze dryer.

### 2.3. Lignin Esterification

Lignin esterification was conducted using the following method [29]. Two grams of DL (3.42 mmol/g of total hydroxyls) was dissolved in THF (15 mL), DMF (2 mL), and pyridine (1.2 mL) at 65 °C for 1 h in the presence of nitrogen gas. Next, 1.71, 3.42, and 5.13 mmol/g of PC, corresponding to 0.5, 1.0, and 1.5 eq./lignin OH, was added using a syringe. The reaction was conducted for 48 h at 65 °C under nitrogen gas with stirring at 150 rpm. EDL was precipitated in water and washed with ethanol to remove unreacted PC in the reactant. After washing with distilled water via vacuum filtration, the obtained EDL was freeze-dried at −55 °C.

### 2.4. Preparation of Composite Film

The DL or EDL were mixed with PLA in ratios of 10/90, 30/70, and 50/50, respectively. The mixtures (0.25 g) were dissolved in chloroform (5 mL) in a shaking incubator (VS-101Si, Vision Scientific Co., Ltd., Republic of Korea) at 40 °C for 6h with stirring at 200 rpm. The dissolved DL/PLA and EDL/PLA solutions in chloroform were poured into PTFE petri dishes and then dried at 25 °C for 12 h. The obtained films from DL/PLA and EDL/PLA were kept in a thermos-hygrostat with a relative humidity of 65%. The thickness of composite films was 0.05–0.08 mm.

### 2.5. Fourier-Transform Infrared Spectroscopy (FTIR)

FTIR analyses were conducted using a Nicolet iS10 (Thermo Fisher Scientific, Waltham, MA, USA) equipped with an attenuated total reflectance attachment. A total of 128 scans were run per sample in the range of 4000–400 cm^−1^.

### 2.6. Gel Permeation Chromatography (GPC)

For GPC analysis, DL or EDL (1 mg) were dissolved in THF (1 mL) and filtered using a syringe filter. The number average molecular weight (Mn), weight average molecular weight (Mw), and polydispersity index (PDI) were measured by GPC (Shimadzu Co., Kyoto, Japan) equipped with ultraviolet (UV) and refractive index detectors. The columns were connected to PLgel 5 μm mixed-C,-D, and PLgel 3 μm mixed-E (Agilent Technologies, Inc., Santa Clara, CA, USA), and the oven temperature was set to 40 °C. The injection volume was 100 μL, and the wavelength of the UV detector was 280 nm. THF was used as the mobile phase under the flow rate of 100 mL/min. A calibration curve was prepared using polystyrene in the range of 1480–1,233,000 g/mol.

### 2.7. Differential Scanning Calorimetry (DSC)

DSC analysis was conducted using a differential scanning calorimeter (SDT Q600, TA instruments, USA) to determine the *T*_g_ of DL and EDL. The samples (5–10 mg) were heated in an aluminum pan under a nitrogen gas purge (100 mL/min). Scanning temperatures ranged from 25–180 °C, with a heating rate of 1 °C/min.

### 2.8. Morphology

Morphologies were observed using a scanning electron microscope (SEM; S-4800, Hitachi, Ltd., Tokyo, Japan) at the Central Laboratory of Kangwon National University. For morphology observation, the surfaces of composite films were coated with iridium to a thickness of 6 nm using a high-vacuum sputter coater (EM ACE600, Leica Microsystems, Ltd., Wetzlar, Germany). The SEM was operated with accelerating voltage of 1 kV at working distance of 8.0–9.0 mm.

### 2.9. Phase Analysis

The phases of composite films was observed using an atomic force microscope (AFM; Nanoscope 5, Bruker, Billerica, MA, USA). The samples for AFM observation were prepared by following method. The dissolved DL and EDL/PLA in chloroform (0.001%) was dropped on the newly cleaved mica disk, and spin-coated using a spin coater (ACE-200, Dong-Ah Trade Corp., Seoul, Korea) at 3000 rpm for 1min. Then, phase imaging was conducted in tapping mode.

### 2.10. Tensile Testing

Specimens were cut from the films according to the Type V dimensions described by the American Society for Testing and Materials D638 standard and maintained in a thermo-hygrostat at 25 °C and a relative humidity of 65% to standardize the effect of relative humidity on the tensile properties. Tensile testing was conducted using a universal testing machine (TO-102, TEST ONE Co., Siheung, Korea) loaded with a load cell of 500 N at a cross-head speed of 10 mm/min. At least nine specimens of each sample were tested and average values obtained.

### 2.11. Water Absorption

The specimens were dried at 60 °C in a vacuum dryer until the weight became constant, and then immersed in a water bath and maintained afterwards at 30 °C. Weight changes were measured periodically using a balance, with a precision of 1 mg, until equilibrium moisture content (*m*_∞_) was attained. Moisture content values against the immersion time (*m*_t_) were calculated using Equation (1):(1)$$m_{t}\left(\%\right)=\frac{W_{t}-W_{o}}{W_{o}}\times 100\left(\%\right)$$
where *W_o_* is the initial weight of the dried sample and *W_t_* is the sample weight against *m_t_* in the water bath.

The diffusion coefficient (D) of water into the composite films can be determined using Fick’s second law [6]. For the case of one-dimensional diffusions from the sheets, Equation (2) was used.
(2)$$\frac{m_{t}}{m_{\infty }}=1-\frac{8}{\pi^{2}}\overset{\infty }{\underset{m=0}{\sum }}\frac{1}{{\left(2m+1\right)}^{2}}exp\left(-\frac{{\left(2m+1\right)}^{2}\pi^{2}Dt}{l^{2}}\right)$$
where *l* is the thickness of the composite film and *t* is the immersion time.

Equation (3) was applied to determine the long-term diffusion coefficient (Dl), where *m_t_*/*m_∞_* > 0.6. The short-term diffusion coefficient (Ds), wherein *m_t_*/*m*_∞_ < 0.6, was determined by Equation (2):(3)$$\frac{m_{t}}{m_{\infty }}=4{\left(\frac{Dt}{\pi l^{2}}\right)}^{\frac{1}{2}}$$

## 3. Results

Table 1 shows the Mn, Mw, and PDI of DL and EDLs prepared at different PC loading. The EDLs had higher Mn and Mw than the DL because of the synthesis of PC on the hydroxyl groups of DL. As PC loading increased, the Mw of EDL increased significantly, indicating a higher PDI. This is because as the amount of PC increases, its synthesized quantity in the hydroxyl group of lignin also increases. Gordobil et al. [30] extracted lignins from spruce and eucalyptus and conducted esterification using dodecanoyl chloride (C12) with DMF, THF and pyridine. The Mws of lignin from spruce and eucalyptus were 3124, 9490 g/mol, respectively. As a result of esterification, the esterified lignin from spruce and eucalyptus present higher molecular weight with higher PDI. It was reported that the carbohydrates chains linked to lignin due to esterification can increase the hydrodynamic volume of lignin and, therefore, increase the apparent molecular weight of the lignin.

Figure 1 shows the FTIR spectra of DL and EDLs prepared at different PC loading. In all the samples, the peaks at 3400 cm^−1^ and 2924 cm^−1^, corresponding to O–H stretching and C–H stretching, were prominent. As the PC loading increased to 1.5 eq./lignin OH, the intensity at 2924 cm^−1^, corresponding to C–H stretching, increased significantly. This is attributed to the esterification of DL with PC. The strong bands in the range of 1710–1760 cm^−1^ were attributed to C=O stretching. The aromatic skeleton vibrations were established at 1602, 1507, and 1414 cm^−1^, which were typical lignin bands. The peak at 1455 cm^−1^ corresponds to C–H deformation in methyl and methylene. The C–O stretching in the guaiacyl ring was evident at 1270 cm^−1^, while the aromatic C–H deformation manifested at 1100–1030 cm^−1^.

Figure 2 indicated the DSC curve of DL and EDL prepared at different PC loading. The *T*_g_ of DL was indicated at 74 °C. It is known that lignin has a relatively high *T*_g_ because condensed rigid phenolic moieties and strong intermolecular hydrogen bonding interactions restrict the thermal mobility of lignin molecules [31]. However, the *T*_g_ of DL tended to be decreased as a result of esterification. As the PC loading increased to 1.5 eq./lignin OH, the *T*_g_ gradually decreased to 50 °C. As a result of esterification with PC, some hydroxyl groups were replaced by ester substituents. This reduced the amount of hydrogen bonding and led to an increased free volume in the molecules and mobility of the chains, resulting in the decreased *T*_g_ [31,32]. Koivu et al. [33] conducted the esterification of kraft lignin using octanoyl chloride (C8), lauroyl chloride (C12), and palmitoyl chloride (C16) under DMF, THF and pyridine. The *T*_g_ of kraft lignin esters was decreased with the loading content of the fatty acid chlorides (C8, C12, C16).

Figure 3 shows the surface morphologies of the neat PLA, and DL or EDL (1.5)/PLA (50/50) composite films. The neat PLA exhibited a smooth surface. For the DL/PLA (50/50) composite film, the surface was rough with some pores, and DL particles were observed. This can be explained the compatibility between DL or EDL and PLA. It is known that lignin has poor compatibility with polyesters including PLA, polybutylene succinate, polycaprolactone [4]. Therefore, it is a common phenomenon that lignin is immiscible with the PLA matrix. The EDL (1.5)/PLA (50/50) composite film evinced a cleaner surface than the DL/PLA (50/50) composite film, indicating no lignin particle and pores. This might be because compatibility between DL and PLA was enhanced due to the esterification. It is known that the esterification on hydroxyl groups can contribute to increased compatibility of lignin with hydrophobic polymers. Maldhure et al. [33] reported the effect of lignin esterification on properties of lignin/polypropylene blends. They observed that the morphological characteristics in the lignin/polypropylene composites could be enhanced as a result of lignin esterification.

Figure 4 shows AFM topography of the surface of neat PLA, DL and EDL (1.5)/PLA (50/50) composite films. Neat PLA presents a very flat surface. Due to the low compatibility between DL and PLA, DL/PLA showed a rougher surface and wave patterns in the phase image. As compared with DL/PLA, the EDL (1.5)/PLA (50/50) had a relatively more uniform phase, indicating smaller patterns. This result was attributed to the enhancement of compatibility between EDL and PLA due to esterification with PC. As a result of esterification with PC, esters with long hydrocarbon chains are formed on the lignin surface, and these can further improve compatibility with PLA.

Figure 5 shows the tensile strength, elastic modulus, and elongation at break of the neat PLA, and DL and EDL (1.5)/PLA composite films with different DL or EDL contents. The neat PLA film had a higher tensile strength than PLA films with lignin. As the DL or EDL contents increased, the tensile strength was decreased. This phenomenon may be due to the weak interfacial adhesion between DL or EDL and PLA in the composite films. It is well known that the tensile strength of lignin-based composites tends to be decreased with increasing lignin content, resulting from poor compatibility with polyesters. The elastic modulus of PLA composite film with DL or EDL was also decreased with increasing content of DL or EDL. Park et al. [6] prepared kraft lignin/PLA composites via twin-screw extrusion with different ratios of lignin and PLA. As kraft lignin loading was increased to 30%, the tensile strength and elastic modulus deteriorated because of weak interfacial bonding between kraft lignin and PLA. EDL/PLA composite films had higher tensile strength and elastic modulus than DL/PLA composite films. This might be because the esterification of DL improved compatibility between lignin and PLA. Increasing hydrophobicity via surface modification of lignin can increase compatibility with other polymers. Yue et al. [34] prepared poly(butylene succinate) (PBS)/wood fiber composite (70/30) via melt compounding at 130 °C and investigated the effect of wood fiber coating with esterified lignin on properties of the composites. The lignin was esterified with octanoyl chloride and PC. It was reported that coating with esterified lignin on wood fiber improves interfacial bonding between wood fiber and PBS, thereby improving mechanical properties of the composites. The elongation at break was decreased with increasing DL or EDL contents. This is because the brittleness in the composites increased as the content of lignin increased.

Figure 6 shows the water absorption behavior of neat PLA, DL and EDL/PLA composite films with different lignin contents. The equilibrium moisture content whereat no further water absorption occurred was determined as the maximum moisture absorption point. In all samples, the water absorption amount increased rapidly for 2 h and then increased at a much lower rate with increasing immersion time in water. The value of maximum water absorption in the neat PLA was approximately 1.8%. The increase of the DL contents in DL/PLA composite films resulted in the increase in the value of maximum water absorption. Although lignin is an aromatic hydrophobic polymer consisting of phenyl propane units, a few lignin hydroxyl groups may cause the water absorption. However, in EDL (1.5)/PLA composite films, water absorption decreased with increasing EDL content. Since esterification with PC occurred at the hydroxyl groups in DL, the hydrophobicity of lignin increased as a result of esterification, resulting in lower water absorption.

From the relationship between the water absorption amount and immersion time shown in Figure 6, the diffusion coefficient of water was determined in Figure 7. The plots of *m_t_*/*m_∞_* versus t_1/2_ and ln(1-*m_t_*/*m_∞_*) versus t were obtained by the Equations (2) and (3), respectively. The short-term and long-term diffusion coefficients were determined from the gradient of the linear regressions. The obtained values are indicated in Table 2.

The Ds and Dl of neat PLA were 10.6 × 10^12^ /m^2^ s^−1^ and 5.0 × 10^12^ /m^2^ s^−1^, respectively. All samples exhibited rapid water absorption in the initial stage, which slowed down after 2 h with increasing immersion time. As a result of DL and EDL addition, the Ds and Dl in the composite decreased. The presence of lignin in the composites may contribute to the inhibition of the rate of water absorption. Moreover, since the esterification on DL increases hydrophobicity [25], lower Ds and Dl were exhibited in EDL/PLA composites. Yue et al. [34] also reported the effect of esterified lignin on water absorption of PBS/wood fiber composites. It was stated that coating with the esterified lignin on wood fiber blocked hydrogen bonding with water, thus the water absorption with the esterified lignin was decreased compared with the composite without the esterified lignin.

## 4. Conclusions

DL was isolated from lignocellulose by DES treatment with ChCl/LA and the DL was esterified with PC. The effect of DL esterification with PC on the morphological characteristics, tensile properties and the thermal and water resistance properties of DL/PLA composites was investigated in this study. As the PC loading increased for DL esterification, the Mw of EDL was increased. DL esterification resulted in the decrement of *T*_g_. The compatibility between DL and PLA could be improved due to esterification; thus, the EDL/PLA composites indicated a higher tensile strength and elastic modulus than the DL/PLA composites. Additionally, DL esterification decreased water absorption with lower water diffusion coefficients. In the development of lignin-based composite materials, the problem of strength deterioration due to the addition of lignin remains to be solved. In this study, it was demonstrated that the esterification of DL using fatty acid could enhance the physico-mechanical properties; this could contribute to the development of high-strength lignin-based composites. Moreover, it is expected that the esterification can contribute to the effective utilization of lignin as various functional materials.

## Figures and Tables

**Figure 1 polymers-13-02149-f001:**
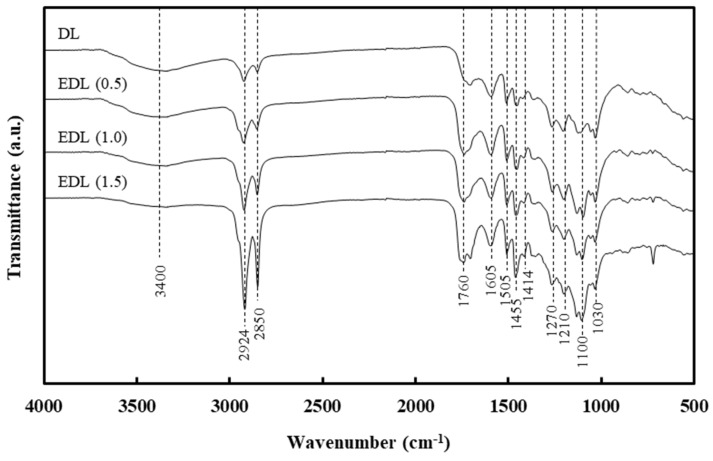
FTIR spectra of DL and EDL prepared at different PC loading.

**Figure 2 polymers-13-02149-f002:**
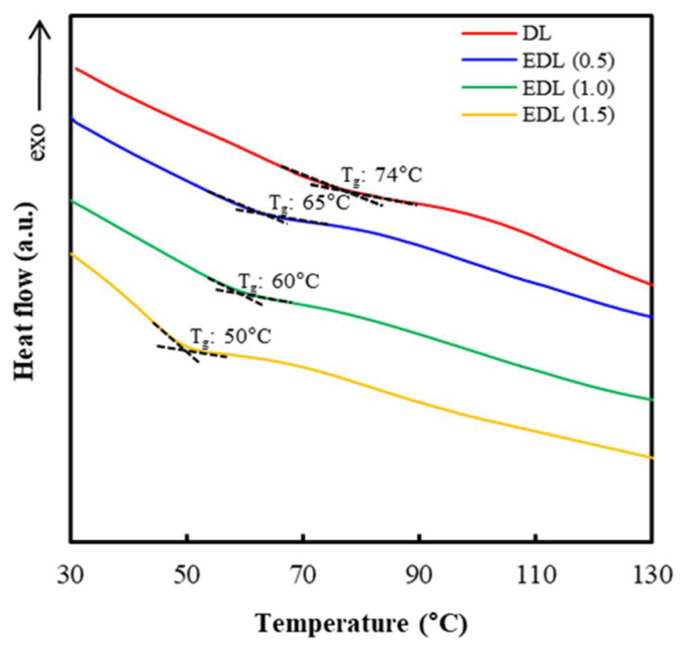
DSC thermograms of DL and EDL prepared at different loading PC.

**Figure 3 polymers-13-02149-f003:**

SEM micrographs of the surface of neat PLA, DL/PLA and EDL (1.5)/PLA (50/50) composite films.

**Figure 4 polymers-13-02149-f004:**
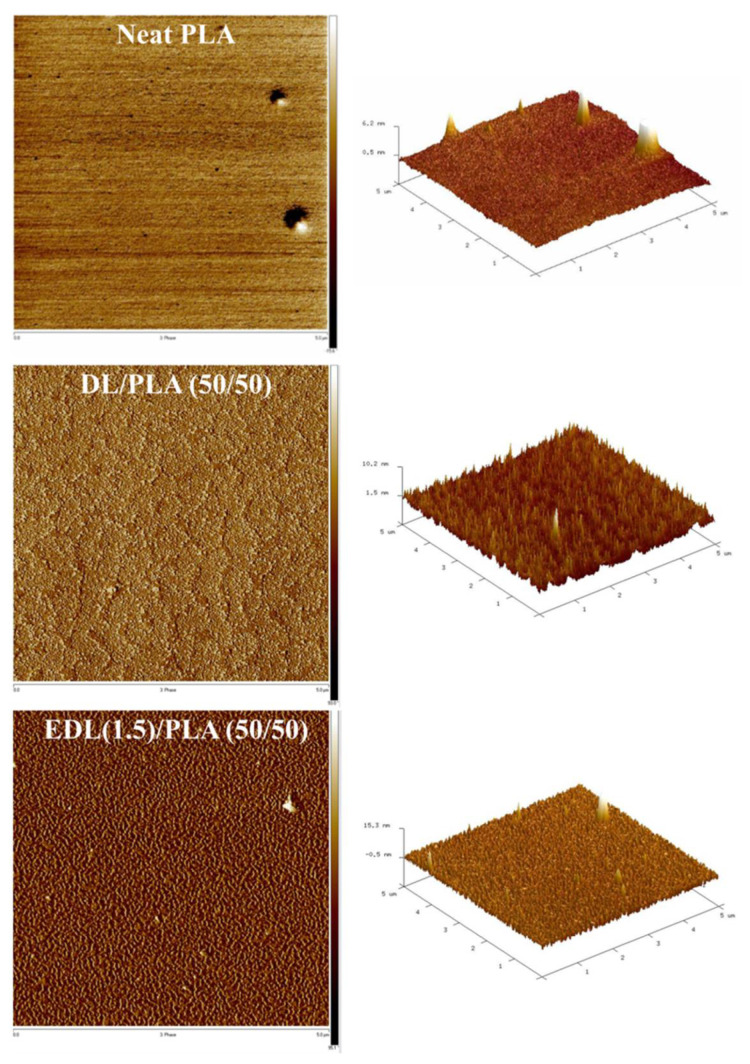
AFM topography of the surface of neat PLA, DL/PLA and EDL (1.5)/PLA (50/50) composite films.

**Figure 5 polymers-13-02149-f005:**
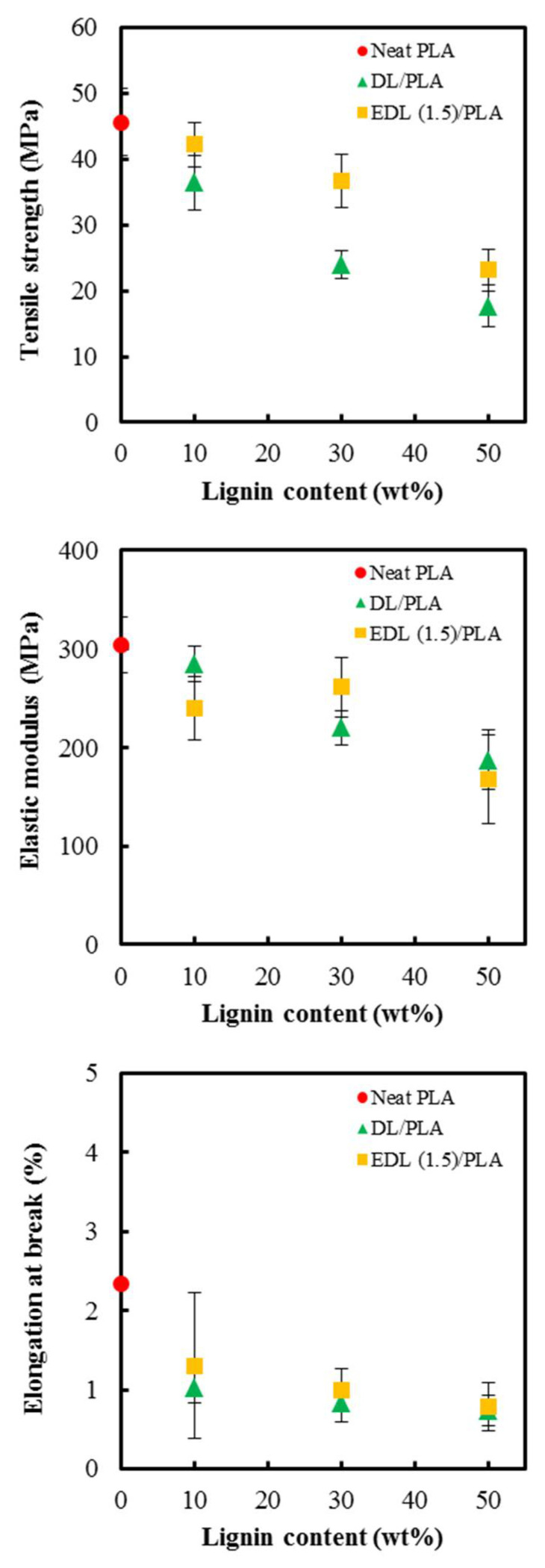
Tensile strength, elastic modulus and elongation at break of neat PLA, and DL/PLA and EDL (1.5)/PLA composite films with different lignin content.

**Figure 6 polymers-13-02149-f006:**
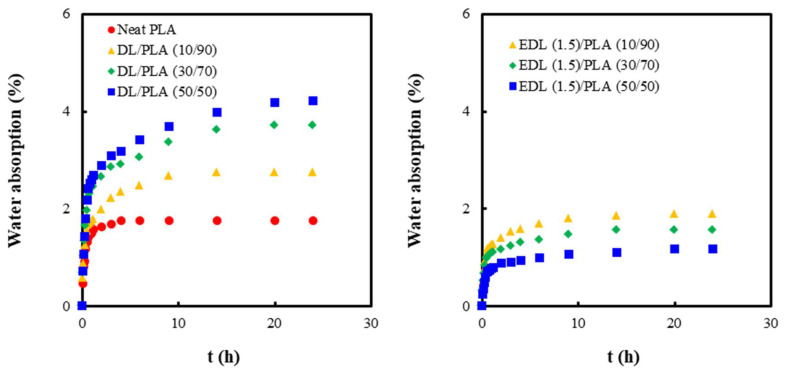
Water absorption curves of neat PLA, DL/PLA and EDL (1.5)/PLA composite films with increasing immersion time in water.

**Figure 7 polymers-13-02149-f007:**
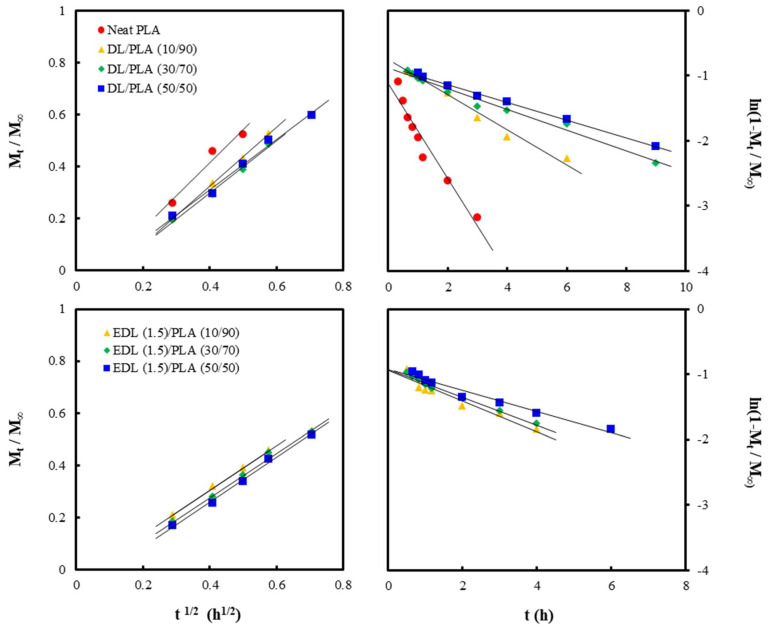
Plots of *mt*/*m∞* versus t1/2 and ln(1-*mt*/*m∞*) versus t from neat PLA, DL/PLA or EDL (1.5)/PLA composite films.

**Table 1 polymers-13-02149-t001:** Mn, Mw, and PDI of DL and EDL prepared at different PC loading.

Sample Code	Loading of PC * (Equivalent/Lignin OH)	Mn (g/mol)	Mw (g/mol)	PDI (Mn/Mw)
DL	-	1943	6580	3.4
EDL(0.5)	0.5	2541	9177	3.6
EDL(1.0)	1.0	2149	11,704	5.5
EDL(1.5)	1.5	2088	13,659	6.5

* PC: palmitoyl chloride.

**Table 2 polymers-13-02149-t002:** Maximum water absorption amount and diffusion coefficient of neat PLA, DL/PLA and EDL (1.5)/PLA composite films containing different lignin contents.

Sample	Maximum Water Absorption Amount (%)	Diffusion Coefficient			
		Short-Term (Ds)		Long-Term (Dl)	
		D*_s_* × 10^12^/m^2^ s^−1^	R^2^	D*_l_* × 10^12^/m^2^ s^−1^	R^2^
Neat PLA	1.8	10.6	0.978	5.0	0.920
DL/PLA (10/90)	2.7	4.6	0.999	2.0	0.993
DL/PLA (30/70)	3.7	3.7	0.990	1.2	0.987
DL/PLA (50/50)	4.2	3.3	0.989	0.9	0.994
EDL(1.5)/PLA (10/90)	1.9	2.1	0.994	1.2	0.927
EDL(1.5)/PLA (30/70)	1.5	3.0	0.995	1.3	0.986
EDL(1.5)/PLA (50/50)	1.2	2.6	0.989	0.9	0.967

## Data Availability

The data presented in this study are available on request from the corresponding author.

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
