# Peer review of "Esterification of Lignin Isolated by Deep Eutectic Solvent Using Fatty Acid Chloride, and Its Composite Film with Poly(lactic acid)"

_polymers, 2021, doi:10.3390/polym13132149_

Round 1
Reviewer 1 Report
Chief Editor
Manuscript ID: Polymers-1268005
Title: Esterification of lignin isolated by deep eutectic solvent using 2 fatty acid chloride, and its composite film with poly (lactic acid)
We are pleased to send you major comments. The studied was done by professionals who deeply understand the essence of the subject. Results seems to be unique. Overall, the presentation style of article is acceptable for Polymers. There are a lot of common mistakes which required major revision for acceptance. I suggest author please put a sincere effort during revision because Polymer is Q1 high reputed journal. To maintain the status of journal it necessary to make the article to polymer standard.
- Please revise the abstract thoroughly because it does not deliver the main theme of the work.
- The key word ``Deep Eutectic Solvent`` should be delete or replace other precise word.
- The introduction should be start from Line 34. The rest pervious Line 28-33 should be delete or transfer to other place.
- Line 48 need a reference or should be rewrite because authors can not say specific things without evidence.
- What is DES? Author should describe each abbreviation in main body before using. Although, this word described in abstract but should be define in main body. Please check thoroughly there are several errors.
- Line 57 to 59. The sentence is too weird and long. Please rewrite it.
- Is it DES or DESs...??
- The novelty of the work must be clearly addressed and discussed, compare your research with existing research findings and highlight novelty, (compare your work with existing research findings and highlight novelty).
- Please don’t use lumpy reference (such as: [6-9] [14-18]. Each reference needs to be properly addressed. Please revise your paper accordingly since same issue occurs on several spots in the paper. The maximum reference in introduction should be 30. Please remove the reference [10-12] and put this single reference only (Preparation and characterization of nanosized lignin from oil palm (Elaeis guineensis) biomass as a novel emulsifying agent).
- Replace the reference [19] by this reference i.e., Thermal degradation and kinetics stability studies of oil palm (Elaeis Guineensis) biomass-derived lignin nanoparticle and its application as an emulsifying agent.
- The main objective of the work must be written on the more clear and more concise way at the end of introduction section.
- Introduction section must be written on more quality way, i.e., more up-to-date references addressed. Research gap should be delivered on more clear way with directed necessity for the conducted research work.
- The arrangement of section 2 is not suitable please use sub heading this in this section. Describe your procedure separately and analysis. Overall, this part is written well.
- I am bit surprised authors did not use any reference in Section 2. Is it new Novel? Please use reference.
- Gordobil et al. (2016) [46] is wrongly written. It should be like Gordobil et al. [46]. I suggest to author please use any published Polymer article for reference in order to prepare the manuscript. There are several errors please correct.
- Is it same or different value 3124, 9490 g/mol?
- Please provide original Figure 3 without editing the image. The original machine image should provide for data verification.
- What is the meaning of surface of neat?
- Conclusion section is missing some perspective related to the future research work, quantify main research findings, highlight relevance of the work with respect to the updated aspect.
- English language should be carefully checked and carefully check paper for language typos, it need major revision.
- Regarding the replications, authors confirmed that replications of experiment were carried out. However, these results are not shown in the manuscript, how many replicated were carried out by experiment? Results seem to be related to a unique experiment. Please, clarify whether the results of this document are from a single experiment or from an average resulting from replications. If replicated were carried out, the use of average data is required as well as the standard deviation in the results and figures shown throughout the manuscript. In case of showing only one replicate explain why only one is shown and include the standard deviations.
Author Response
Reviewer 1
We are pleased to send you major comments. The studied was done by professionals who deeply understand the essence of the subject. Results seems to be unique. Overall, the presentation style of article is acceptable for Polymers. There are a lot of common mistakes which required major revision for acceptance. I suggest author please put a sincere effort during revision because Polymer is Q1 high reputed journal. To maintain the status of journal it necessary to make the article to polymer standard.
Comment 1
Please revise the abstract thoroughly because it does not deliver the main theme of the work.
Answer 1
Thank you for your valuable review. We have revised the abstract to better convey the purpose of this study.
Comment 2
The key word ``Deep Eutectic Solvent`` should be delete or replace other precise word.
Answer 2
We’ve replace the key word ‘Deep eutectic solvent’ to ‘composite’. Thank you
Comment 3
The introduction should be start from Line 34. The rest pervious Line 28-33 should be delete or transfer to other place.
Answer 3
Thank you for your critical review. To focus more on the content of lignin, the line 28-33 have been deleted.
Comment 4
Line 48 need a reference or should be rewrite because authors can not say specific things without evidence.
Answer 4
We’ve deleted the sentence in line 48. Thank you
Comment 5
What is DES? Author should describe each abbreviation in main body before using. Although, this word described in abstract but should be define in main body. Please check thoroughly there are several errors.
Answer 5
Thank you so much. We’ve checked and corrected all typos in this manuscript thoroughly.
Comment 6
Line 57 to 59. The sentence is too weird and long. Please rewrite it.
Answer 6
We’ve revised the sentence to help readers understand. Thank you.
Comment 7
Is it DES or DESs...??
Answer 7
We’ve changed the word DESs to DES.
Comment 8
The novelty of the work must be clearly addressed and discussed, compare your research with existing research findings and highlight novelty, (compare your work with existing research findings and highlight novelty).
Answer 8
We agree with you that it is necessary to emphasize the novelty of our study. Therefore, we have revised the manuscript to emphasize the novelty of our research to our readers.
Comment 9
Please don’t use lumpy reference (such as: [6-9] [14-18]. Each reference needs to be properly addressed. Please revise your paper accordingly since same issue occurs on several spots in the paper. The maximum reference in introduction should be 30. Please remove the reference [10-12] and put this single reference only (Preparation and characterization of nanosized lignin from oil palm (Elaeis guineensis) biomass as a novel emulsifying agent).
Answer 9
References 10-11 have been deleted, and the references you mentioned have been added.
Comment 10
Replace the reference [19] by this reference i.e., Thermal degradation and kinetics stability studies of oil palm (Elaeis Guineensis) biomass-derived lignin nanoparticle and its application as an emulsifying agent.
Answer 10
The references you mentioned have been added. The reference number is [9]. Thank you.
Comment 11
The main objective of the work must be written on the more clear and more concise way at the end of introduction section.
Answer 11
Thank you for your critical review. We’ve modified the manuscript in the end of Introduction to emphasize the main objective of this work.
Comment 12
Introduction section must be written on more quality way, i.e., more up-to-date references addressed. Research gap should be delivered on more clear way with directed necessity for the conducted research work.
Answer 12
In previous work, we tried to lignin plasticization for improvement of physico-mechanical properties of lignin-based biodegradable composite. In that study, it was confirmed that the improvement of interfacial adhesion between lignin and polymer could improve the physical properties of the composite material. However, in order to improve the strength of the lignin composite material, improvement of compatibility between lignin and polymer should be prioritized. Therefore, in this study, esterification was performed to improve compatibility with PLA. The importance of improving compatibility is included in the introduction. Thank you so much.
Comment 13
The arrangement of section 2 is not suitable please use sub heading this in this section. Describe your procedure separately and analysis. Overall, this part is written well.
Answer 13
Thank you for your comment. We have modified the manuscript to have sub heading in Experiment part.
Comment 14
I am bit surprised authors did not use any reference in Section 2. Is it new Novel? Please use reference.
Answer 14
We modified section 2 to include references. Thank you for your comment.
Comment 15
Gordobil et al. (2016) [46] is wrongly written. It should be like Gordobil et al. [46]. I suggest to author please use any published Polymer article for reference in order to prepare the manuscript. There are several errors please correct.
Answer 15
Thank you. All modifications have been made according to the policy of Polymers.
Comment 16
Is it same or different value 3124, 9490 g/mol?
Answer 16
Mw of lignin from spruce is 3124 g/mol, and Mw of lignin from eucalyptus is 9490 g/mol. Thank you.
Comment 17
Please provide original Figure 3 without editing the image. The original machine image should provide for data verification.
Answer 17
Thank you for your review. The result in Fig 3 has been changed to original data.
Comment 18
What is the meaning of surface of neat?
Answer 18
We have used the word ‘neat PLA’ to express PLA without fillers. The ‘surface of neat PLA’ means the morphological characteristic of ‘neat PLA’ on its surface. Thank you.
Comment 19
Conclusion section is missing some perspective related to the future research work, quantify main research findings, highlight relevance of the work with respect to the updated aspect.
Answer 19
Thank you for your critical review. Based on your advice, we’ve revised the conclusion section. Thank you.
Comment 20
English language should be carefully checked and carefully check paper for language typos, it need major revision.
Answer 20
Thank you for your review. This manuscript has been reviewed by a native speaking professional editor. We have checked language again.
Comment 21
Regarding the replications, authors confirmed that replications of experiment were carried out. However, these results are not shown in the manuscript, how many replicated were carried out by experiment? Results seem to be related to a unique experiment. Please, clarify whether the results of this document are from a single experiment or from an average resulting from replications. If replicated were carried out, the use of average data is required as well as the standard deviation in the results and figures shown throughout the manuscript. In case of showing only one replicate explain why only one is shown and include the standard deviations.
Answer 21
Thank you for your comment. The Tensile testing of the film was repeated to calculate the average value and standard deviation, which can be seen in Fig.5. Nine specimens were tested and the average value was calculated, which is specified in section 2.
Reviewer 2 Report
Generally, the paper present the interesting way the increase lignin compatibility with PLA and improve the properties of the final composite for the potential usage.
Some remarks are below:
lines 30-32: It is not obvious that biodegradation of materials means that they are ecological.
line 34: Petroleum-based plastic does not mean dengerous for the environment. Sometimes they are more friendly for environment that biodegradable ones. They have e.g. a good energetic balance.
line 64-69: Please, specify what thermoplastic polymers the Authors mean.
Introduction, general remark: Please, write claerly what is the research hypothesis and what is new and what is new to the present knowledge.
Experimental: The used solvents e.g. THF, DMF are not ecological. Did Author take into account maybe some other solvents or the way of recovering them that they will be not emitted to the environment?
Author Response
Reviewer 2
Generally, the paper present the interesting way the increase lignin compatibility with PLA and improve the properties of the final composite for the potential usage.
Some remarks are below:
Comment 1
lines 30-32: It is not obvious that biodegradation of materials means that they are ecological.
Answer 1
Thank you so much for your valuable review. In order to emphasize the lignin and lignin esterification, the paragraph in lines 30-32 was deleted.
Comment 2
line 34: Petroleum-based plastic does not mean dengerous for the environment. Sometimes they are more friendly for environment that biodegradable ones. They have e.g. a good energetic balance.
Answer 2
The manuscript has been modified to start with the lignin. Thank you so much.
Comment 3
line 64-69: Please, specify what thermoplastic polymers the Authors mean.
Answer 3
We have modified it to specify the thermoplastic polymer. Thank you so much.
Comment 4
Introduction, general remark: Please, write claerly what is the research hypothesis and what is new and what is new to the present knowledge.
Answer 4
Thank you for your valuable comments. The main hypothesis of the study is that the synthesis of fatty acids to lignin may enhance compatibility with PLA. The manuscript has been revised to emphasize its novelty. Thank you.
Comment 5
Experimental: The used solvents e.g. THF, DMF are not ecological. Did Author take into account maybe some other solvents or the way of recovering them that they will be not emitted to the environment?
Answer 5
We have agreed that the solvents we used are not ecological. The used DMF and THF are discarded as wastewater. Our research team has focused on establishing green chemical processes. We are developing a way for efficient esterification of lignin without THF and DMF. It is almost in the stage of success, and the research results will be published. Thank you so much.
Reviewer 3 Report
Dear Author
In this paper, the authors reported on the Esterification of lignin isolated by deep eutectic solvent using 2 fatty acid chloride, and its composite film with poly(lactic acid).Through the application of various analytical procedures, the authors produced many interesting results. I think the water resistance is a reasonable value, but EDL1.5 alone is not enough for mechanical properties, and mechanical properties of 0.5 and 1.0 are also required. However, there are also some problems, which the authors need to address by answering the following questions clearly.
- What is the yield of regenerated DL after treatment with lignin using the ionic liquid EDS?
- Page4 Line154 This word (filmss) is misspelled. 「The diffusion coefficient (D) of water into the composite filmss can be determined using Fick’s second law.」
- Regarding the chemical structure of EDL, what is the yield and degree of substitution (DS) of EDL after esterification of DL?
- What are the particle sizes of DL and EDL?
- The thickness of DL / PLA and EDL / PLA films used in SEM and AFM is required for this paper.
- results Here, polylactic acid and DL or EDL are added to evaluate the mechanical properties, but chemical bonds (ester bonds, hydrogen bonds, etc.) need to be explained in detail. Alternatively, it is necessary to show a possible chemical structure.
- If you change the scale of water absorption in Figure 6 from MAX10 to 6, the plot will be easier to see.
Author Response
Reviewer 3
In this paper, the authors reported on the Esterification of lignin isolated by deep eutectic solvent using 2 fatty acid chloride, and its composite film with poly(lactic acid).Through the application of various analytical procedures, the authors produced many interesting results. I think the water resistance is a reasonable value, but EDL1.5 alone is not enough for mechanical properties, and mechanical properties of 0.5 and 1.0 are also required. However, there are also some problems, which the authors need to address by answering the following questions clearly.
Comment 1
What is the yield of regenerated DL after treatment with lignin using the ionic liquid EDS?
Answer 1
Thank you for your valuable review. When lignocellulose was treated with ChCl/LA at 130°C for 24 hours, than about 90% of the lignin in lignocellulosic was isolated. The yield and characteristics of lignin dependent on reaction temperature and time were already analyzed and will be submitted to the journal.
Comment 2
Page4 Line154 This word (filmss) is misspelled. 「The diffusion coefficient (D) of water into the composite filmss can be determined using Fick’s second law.」
Answer 2
Thank you so much. We have checked the words in this manuscript carefully.
Comment 3
Regarding the chemical structure of EDL, what is the yield and degree of substitution (DS) of EDL after esterification of DL?
Answer 3
Thank you for your valuable comment. Degree of substitution is carried out by comparing the hydroxyl content of lignin. We have a plan to submit a research paper focusing on the esterification of lignin soon.
We have a plan to publish
Comment 4
What are the particle sizes of DL and EDL?
Answer 4
Spherical lignin particles are observed in the DL/PLA (50/50) film, and their size is observed to be 1-3um. However, the particles of lignin are not well observed in the EDL (1.4)/PLA (50/50) sample.
Comment 5
The thickness of DL / PLA and EDL / PLA films used in SEM and AFM is required for this paper.
Answer 5
Thank you for your review. We have added the information on the thickness of films. Thank you.
Comment 6
Here, polylactic acid and DL or EDL are added to evaluate the mechanical properties, but chemical bonds (ester bonds, hydrogen bonds, etc.) need to be explained in detail. Alternatively, it is necessary to show a possible chemical structure.
Answer 6
Thank you for your comment. The interaction between lignin and PLA is very important in mechanical properties. In previous study, we investigated the effect of coupling agent on lignin/PLA composites. As pMDI as a coupling agent was added in the composite, the urethane bonds occurred between lignin and PLA, it contribute to improve the mechanical properties. This study only focused the compatibility and mixability between lignin and PLA. We hypothesized that esterified lignin has better compatibility with polyester PLA. Therefore, based on the results of compatibility presented in AFM and SEM, the results of tensile properties were considered.
Comment 7
If you change the scale of water absorption in Figure 6 from MAX10 to 6, the plot will be easier to see.
Answer 7
We have modified the scale of Y-axis from 10 to 6. Thank you for your comments.